# SHARPER UTILITY BOUNDS FOR DIFFERENTIALLY PRIVATE MODELS

## ABSTRACT

In this paper, by introducing Generalized Bernstein condition, we propose the first $\mathcal{O}\big(\frac{\sqrt{p}}{n\epsilon}\big)$ high probability excess population risk bound for differentially private algorithms under the assumptions $G$-Lipschitz, $L$-smooth, and Polyak-Łojasiewicz condition, based on gradient perturbation method. If we replace the properties $G$-Lipschitz and $L$-smooth by $\alpha$-Hölder smoothness (which can be used in non-smooth setting), the high probability bound comes to $\mathcal{O}\big(n^{-\frac{2\alpha}{1+2\alpha}}\big)$ w.r.t $n$, which cannot achieve $\mathcal{O}\left(1/n\right)$ when $\alpha \in (0,1]$. To solve this problem, we propose a variant of gradient perturbation method, **max**$\{1,g\}$**-Normalized Gradient Perturbation** (m-NGP). We further show that by normalization, the high probability excess population risk bound under assumptions $\alpha$-Hölder smooth and Polyak-Łojasiewicz condition can achieve $\mathcal{O}\big(\frac{\sqrt{p}}{n\epsilon}\big)$, which is the first $\mathcal{O}\left(1/n\right)$ high probability utility bound w.r.t $n$ for differentially private algorithms under non-smooth conditions. Moreover, we evaluate the performance of the new proposed algorithm m-NGP, the experimental results show that m-NGP improves the performance (measured by accuracy) of the DP model over real datasets. It demonstrates that m-NGP improves the excess population risk bound and the accuracy of the DP model on real datasets simultaneously.

## 1 INTRODUCTION

Machine learning has been widely used and found effective in many fields in recent years (Singha et al., 2021; Swapna & Soman, 2021; Ponnusamy et al., 2021). When training machine learning models, tremendous data was collected, and the data often contains sensitive information of individuals, which may leakage personal privacy (Shokri et al., 2017; Carlini et al., 2019).

Differential Privacy (DP) (Dwork et al., 2006; Dwork & Lei, 2009; Dwork et al., 2014) is a theoretically rigorous tool to prevent sensitive information. It introduces random noise to the machine learning model and blocks adversaries from inferring any single individual included in the dataset by observing the model. The mathematical definition of DP is well accepted and relative technologies are performed by Google (Erlingsson et al., 2014), Apple (McMillan, 2016) and Microsoft (Ding et al., 2017). As such, DP has attracted attention from the researchers and has been applied to numerous machine learning problems (Ullman & Sealfon, 2019; Xu et al., 2019; Bernstein & Sheldon, 2019; Wang & Xu, 2019; Heikkilä et al., 2019; Kulkarni et al., 2021; Bun et al., 2021; Nguyen & Vullikanti, 2021).

There are mainly three approaches to guarantee differential privacy: output perturbation (Chaudhuri et al., 2011), objective perturbation (Chaudhuri et al., 2011), and gradient perturbation (Song et al., 2013). Considering that gradient descent is a widely used optimization method, the gradient perturbation method can be used for a wide range of applications, and adding random noise to the gradient allows the model to escape local minima (Raginsky et al., 2017), we focus on the gradient perturbation method to guarantee DP in this paper.

In this paper, we aim to minimize the population risk, and measure the utility of the DP model by the excess population risk. To get the excess population risk, an important step is to analyze the generalization error (the reason is demonstrated in Section 3). Complexity theory (Bartlett et al., 2002) and algorithm stability theory (Bousquet & Elisseeff, 2002) are popular tools to analyze the generalization error. On one hand, Chaudhuri et al. (2011) applied the complexity theory and achieved

an $\mathcal{O}\big(\max\{\frac{1}{\sqrt{n}}, \sqrt[2/3]{\frac{p}{n\epsilon}}\}\big)$ high probability excess population risk bound under the assumption of strongly convex; Kifer et al. (2012) achieved $\mathcal{O}\big(\frac{\sqrt{p}}{n\epsilon}\big)$ expected excess population risk bound via complexity theory. On the other hand, the sharpest known high probability generalization bounds for DP algorithms analyzed via stability theory under different assumptions (Wu et al., 2017; Bassily et al., 2019; Feldman et al., 2020; Bassily et al., 2020; Wang et al., 2021) are $\mathcal{O}\big(\frac{\sqrt{p}}{n\epsilon} + \frac{1}{\sqrt{n}}\big)$ or $\mathcal{O}\big(\frac{\sqrt[4]{p}}{\sqrt{n\epsilon}}\big)$, containing an inevitable $\mathcal{O}\big(\frac{1}{\sqrt{n}}\big)$ term, which is a bottleneck on the utility analysis. Thus, we are focusing on the following question, which is still an open problem:

*Can we achieve the high probability excess risk bounds with rate $\mathcal{O}(\frac{\sqrt{p}}{n\epsilon})$ for differentially private models via uniform stability?*

This paper answers the question positively under more (or different) assumptions and provides the first high probability bound allowing an $\mathcal{O}\big(\frac{\sqrt{p}}{n\epsilon}\big)$ rate of convergence in the setting of DP. By introducing *Generalized Bernstein condition* (Koltchinskii, 2006), we remove the $\mathcal{O}\big(\frac{1}{\sqrt{n}}\big)$ term in the generalization error and furthermore improve the high probability excess population risk bound. Comparing with previous high probability bounds, the improvement is approximately up to $\mathcal{O}\left(\sqrt{n}\right)$.

### CONTRIBUTIONS

We first prove that by introducing Generalized Bernstein condition (Koltchinskii, 2006), under the assumptions $G$-Lipschitz, $L$-smooth, and Polyak-Łojasiewicz (PL) condition, the high probability excess population risk bound can be improved to $\mathcal{O}\big(\frac{\sqrt{p}}{n\epsilon}\big)$. To the best of our knowledge, this is the first $\mathcal{O}\big(\frac{\sqrt{p}}{n\epsilon}\big)$ high probability excess population risk bound in the field of DP.

Then, we relax the assumptions $G$-Lipschitz and $L$-smooth, by introducing $\alpha$-Hölder smooth. Under these assumptions, we prove that the high probability excess population risk bound comes to $\mathcal{O}\big(\frac{\sqrt{p}}{\epsilon} n^{\frac{-2\alpha}{1+2\alpha}}\big)$. Considering that $\alpha \in (0, 1]$, the result cannot achieve $\mathcal{O}\big(\frac{\sqrt{p}}{n\epsilon}\big)$.

To overcome the bottleneck, we design a variant of gradient perturbation method, called **max $\{1, g\}$-Normalized Gradient Perturbation** (m-NGP) algorithm. Via this new proposed algorithm, we prove that under the assumptions $\alpha$-Hölder smooth, PL condition, and generalized Bernstein condition, the high probability excess population risk bound can be improved to $\mathcal{O}\big(\frac{\sqrt{p}}{n\epsilon}\big)$. To the best of our knowledge, this is the first $\mathcal{O}\big(\frac{\sqrt{p}}{n\epsilon}\big)$ high probability excess population risk bound for non-smooth loss in the field of DP.

Moreover, to evaluate the performance of our proposed max $\{1, g\}$-Normalized Gradient Perturbation algorithm, we perform experiments on real datasets, the experimental results show that m-NGP method also improves the accuracy of the DP model on real datasets.

The rest of the paper is organized as follows. We discuss some related work in Section 2. Some preliminaries are formally introduced in Section 3. In Section 4, we propose sharper utility bounds under different assumptions and design a variant of gradient perturbation method, **max $\{1, g\}$-Normalized Gradient Perturbation**. The experimental results are shown in Section 5. Finally, we conclude the paper in Section 6.

## 2 RELATED WORK

Dwork et al. (2006) proposed the mathematical definition of DP for the first time. Then, it was developed to protect the privacy in the field of machine learning (e.g. Empirical Risk Minimization (ERM)) via output perturbation, objective perturbation, and gradient perturbation methods. For DP-ERM formulations, Chaudhuri et al. (2011) first proposed output perturbation and objective perturbation methods, and Song et al. (2013) first proposed the gradient perturbation method. Based on these works, Kifer et al. (2012); Bassily et al. (2014); Abadi et al. (2016); Wang et al. (2017); Zhang et al. (2017); Wu et al. (2017); Bassily et al. (2019); Feldman et al. (2020); Bassily et al. (2020) further improved the results under different assumptions.

Table 1: Previous excess population risk bounds and ours under different assumptions

| | Assumptions | Method | Utility Bound |
|---|---|---|---|
| Bassily et al. (2019) | Lipschitz, smooth, convex | Gradient | $\mathcal{O}\left(\frac{\sqrt{p}}{n\epsilon} + \frac{1}{\sqrt{n}}\right)$ |
| Feldman et al. (2020) | Lipschitz, convex | Gradient | $\mathcal{O}\left(\frac{\sqrt{p}}{n\epsilon} + \frac{1}{\sqrt{n}}\right)$ |
| Bassily et al. (2020) | Lipschitz, convex | Gradient | $\mathcal{O}\left(\frac{\sqrt{p}}{n\epsilon} + \frac{1}{\sqrt{n}}\right)$ |
| Wang et al. (2021) | $\alpha$-Hölder smooth, convex | Gradient | $\mathcal{O}\left(\frac{\sqrt{p}}{n\epsilon} + \frac{1}{\sqrt{n}}\right)$ |
| Wang et al. (2021) | $\alpha$-Hölder smooth, convex | Output | $\mathcal{O}\left(\frac{\sqrt[4]{p}}{\sqrt{n}\epsilon}\right)$ |
| Ours | Lipschitz, smooth, PL condition | Gradient | $\mathcal{O}\left(\frac{\sqrt{p}}{n\epsilon}\right)$ |
| Ours | $\alpha$-Hölder smooth, PL condition | Gradient | $\mathcal{O}\left(\frac{\sqrt{p}}{n^{\frac{2\alpha}{1+2\alpha}}\epsilon}\right)$ |
| Ours (m-NGP) | $\alpha$-Hölder smooth, PL condition | Gradient | $\mathcal{O}\left(\frac{\sqrt{p}}{n\epsilon}\right)$ |

[1] In Table 1, $n$ is the size of the dataset, $\epsilon$ is the privacy budget, and $p$ is the dimension of the data.

Among the works mentioned above, some of them only analyzed the privacy guarantees (Song et al., 2013; Abadi et al., 2016), some of them only discussed the excess empirical risk bound (Wang et al., 2017; Zhang et al., 2017; Wu et al., 2017). Some works discussed the excess population risk under expectation, from different points of view, such as complexity theory, optimization theory, and stability theory: Kifer et al. (2012) achieved an $\mathcal{O}\left(\frac{\sqrt{p}}{n\epsilon}\right)$ expected excess population risk bound via complexity theory; Bassily et al. (2014) achieved similar expected bound under convexity assumption, via optimization theory; and Wang et al. (2019) proposed an $\mathcal{O}\left(\frac{p}{\log(n)\epsilon^2}\right)$ expected excess population risk bound under non-convex condition, via Langevin Dynamics (Gelfand & Mitter, 1991) and the stability of Gibbs algorithm.

Considering that the high probability bound is more concerned by researchers, we focus on the high probability utility bound. Meanwhile, we concentrate on the stability theory in this paper. Among many notions of stability, uniform stability is arguably the most popular one, which yields exponential generalization bounds. Via uniform stability, the high probability excess population risk bounds under different assumptions given by previous works all contain an $\mathcal{O}\left(\frac{1}{\sqrt{n}}\right)$ term, details can be found in Table 1. The reason is that when analyzing the generalization error, the technical routes followed works Bousquet & Elisseeff (2002); Hardt et al. (2016).

In this paper, by introducing *Generalized Bernstein condition* (Koltchinskii, 2006), we remove the $\mathcal{O}\left(\frac{1}{\sqrt{n}}\right)$ term from the generalization error, and further improve the excess population risk bound of DP models. The improved convergence rate is up to $\mathcal{O}\left(\frac{\sqrt{p}}{n\epsilon}\right)$, which positively answers the question: Can the high probability excess population risk bound achieve $\mathcal{O}\left(1/n\right)$ w.r.t $n$. The improvements are shown in Table 1.

Table 1 first shows that by adding more assumptions (we assume the loss function to be Lipschitz, smooth, and satisfy Polyak-Łojasiewicz (PL) condition, while previous results require $\alpha$-Hölder smoothness and convexity), we achieve a better high probability excess population risk bound, $\mathcal{O}\left(\frac{\sqrt{p}}{n\epsilon}\right)$, which is state-of-the-art to the best of our knowledge. Then, we replace the Lipschitz and smooth property by $\alpha$-Hölder smoothness and achieve $\mathcal{O}\left(\frac{\sqrt{p}}{n^{\frac{2\alpha}{1+2\alpha}}\epsilon}\right)$ high probability excess population risk bound, when $\alpha \in [\frac{1}{2}, 1]$, our result is better than previous ones, but it cannot achieve

the same bound ($\mathcal{O}\left(1/n\right)$ w.r.t $n$) under the condition that the loss function is Lipschitz, smooth, and satisfies PL condition. To overcome it, we propose an algorithm called m-NGP, and achieve the $\mathcal{O}\left(\frac{\sqrt{p}}{n\epsilon}\right)$ result under the same assumptions: $\alpha$-Hölder smooth and PL condition.

Moreover, although it is hard to directly compare PL condition with convexity, PL condition can be applied to many non-convex conditions (more information can be found in Section 4.2). So, in this paper, we analyze the utility bound of DP algorithm under cases different from previous scenarios.

## 3 PRELIMINARIES

In this paper, we assume that there are $n$ data instances in dataset $D$, i.e. $D = \{z_1, \cdots, z_n\}$ where $z = (x, y)$ with input $x \in \mathcal{X}$ and label $y \in \mathcal{Y}$, and $\mathcal{Z} = \mathcal{X} \times \mathcal{Y}$. The data space is denoted by $\mathcal{D}$ and the parameter space is denoted by $\mathcal{C}$, the loss function $\ell$ is defined as $\ell(\cdot, \cdot) : \mathcal{D} \times \mathcal{C} \to \mathbb{R}$. Databases $D, D' \in \mathcal{D}^n$ differing by one data instance are denoted as $D \sim D'$, called *adjacent databases*. For a given vector $\boldsymbol{x} = [x_1, \cdots, x_d]^T$, its $\ell_2$-norm is $\|\boldsymbol{x}\|_2 = (\sum_{i=1}^{d} |x_i|^2)^{\frac{1}{2}}$. And $A \lesssim B$ represents that there exists $c > 0$, $A \leq cB$.

**Definition 1** (Differential Privacy (Dwork et al., 2006)). *A randomized algorithm: $\mathcal{A} : \mathcal{D}^n \to \mathbb{R}^p$ is $(\epsilon, \delta)$-differential privacy (DP) if for all $D \sim D'$ and events $S \in range(\mathcal{A})$:*

$$\mathbb{P}\left[\mathcal{A}(D) \in S\right] \leq e^{\epsilon}\mathbb{P}\left[\mathcal{A}(D') \in S\right] + \delta.$$

Definition 1 implies that the adversaries cannot infer whether an individual participates when training the machine learning model, because essentially the same distributions will be drawn over any adjacent datasets. Some kind of attacks, such as membership inference attack, attribute inference attack, and memorization attack, can be thwarted by DP (Backes et al., 2016; Jayaraman & Evans, 2019; Carlini et al., 2019).

Throughout this paper, we focus on gradient perturbation method to guarantee $(\epsilon, \delta)$-DP, the paradigm is based on gradient descent: at iteration $t$,

$$\hat{\theta}_t \leftarrow \hat{\theta}_{t-1} - \eta_t \left(\nabla_\theta R_n(\hat{\theta}_{t-1}) + b\right), \tag{1}$$

where $\eta_t$ is the learning rate, $b$ is the random noise injected into the gradient, $\hat{\theta}$ is corresponding model with privacy, and $R_n(\theta)$ is the empirical risk, defined as $R_n(\theta) := \frac{1}{n}\sum_{i=1}^{n}\ell(z_i, \theta)$.

In this paper, we focus on minimizing the population risk: $R(\theta) = \mathbb{E}_{z \sim \mathcal{D}}\left[\ell(z, \theta)\right]$. In the setting of DP, the excess population risk is defined by $R(\hat{\theta}) - \min_{\theta \in \mathcal{C}} R(\theta)$, which can be decomposed into:

$$R(\hat{\theta}_n) - R(\theta^*) = R(\hat{\theta}_n) - R_n(\hat{\theta}_n) + R_n(\hat{\theta}_n) - R_n(\theta^*) + R_n(\theta^*) - R(\theta^*)$$
$$\leq \underbrace{R(\hat{\theta}_n) - R_n(\hat{\theta}_n)}_{\text{GE}} + \underbrace{R_n(\hat{\theta}_n) - R_n(\theta^*_{\mathbf{n}})}_{\text{OE}} + R_n(\theta^*) - R(\theta^*), \tag{2}$$

where $\theta^* = \arg\min_{\theta \in \mathcal{C}} R(\theta), \theta^*_n = \arg\min_{\theta \in \mathcal{C}} R_n(\theta)$, and the last inequality is becasuse of the definition of $\theta^*_n$. In (2), GE, OE mean the generalization error and the optimization error (also called the excess empirical risk), respectively. Inequality (2) answers the question mentioned in Section 1: Why generalization error is an important step towards excess population risk.

To get the generalization error, algorithm stability theory is a popular tool, in which uniform stability yields exponential generalization bounds and is commonly used.

**Definition 2** (Uniform Stability (Bousquet & Elisseeff, 2002)). *An algorithm $\theta_n$ is $\gamma$-uniformly stable if for any $z, z_1, \cdots, z_i, \cdots, z_n, z'_i \in \mathcal{Z}$ and $i = 1, \cdots, n$, it holds that*

$$|\ell(z, \theta_n(z_1, \cdots, z_n)) - \ell(z, \theta_n(z_1, \cdots, z_{i-1}, z'_i, z_{i+1}, \cdots, z_n))| \leq \gamma.$$

In this paper, we use notation $\theta_n$ for both algorithm and model parameter. By Definition 2, it is easy to follow that the uniform stability measures the upper bound of the difference (on the loss function) between the models derived from adjacent datasets.

**Assumption 1** ($G$-Lipschitz). *The loss function $\ell : \mathcal{D} \times \mathcal{C} \to \mathbb{R}$ is $G$-Lipschitz over $\theta$ if for any $z \in \mathcal{D}$ and $\theta_1, \theta_2 \in \mathcal{C}$, we have: $|\ell(z, \theta_1) - \ell(z, \theta_2)| \leq G\|\theta_1 - \theta_2\|_2$.*

**Assumption 2** (*L*-smooth). *The loss function $\ell : \mathcal{D} \times \mathcal{C} \to \mathbb{R}$ is $L$-smooth over $\theta$ if for any $z \in \mathcal{D}$ and $\theta_1, \theta_2 \in \mathcal{C}$, we have: $\|\nabla_\theta \ell(z, \theta_1) - \nabla_\theta \ell(z, \theta_2)\|_2 \le L\|\theta_1 - \theta_2\|_2$.*

If $\ell$ is differentiable, smoothness yields: $\ell(z, \theta_1) - \ell(z, \theta_2) \le \langle \nabla_\theta \ell(z, \theta_2), \theta_1 - \theta_2 \rangle + \frac{L}{2} \|\theta_1 - \theta_2\|_2^2$.

Assumptions $G$-Lipschitz and $L$-smooth are commonly used in the utility analysis of DP machine learning (Chaudhuri et al., 2011; Kifer et al., 2012; Abadi et al., 2016; Bassily et al., 2019; Feldman et al., 2020; Bassily et al., 2020). To relax the Lipschitz and smoothness assumptions, we introduce the $\alpha$-Hölder smoothness of the loss function:

**Assumption 3** ($\alpha$-Hölder smooth). *Let $\alpha \in (0, 1]$. The loss function $\ell : \mathcal{D} \times \mathcal{C} \to \mathbb{R}$ is $\alpha$-Hölder smooth over $\theta$ with parameter $H$ if for any $z \in \mathcal{D}$ and $\theta_1, \theta_2 \in \mathcal{C}$, we have: $\|\nabla_\theta \ell(z, \theta_1) - \nabla_\theta \ell(z, \theta_2)\|_2 \le H\|\theta_1 - \theta_2\|_2^\alpha$.*

**Lemma 1.** *If the loss function $\ell(\cdot, \cdot)$ is differentiable, then Assumption 3 yields $\ell(z, \theta_1) - \ell(z, \theta_2) \le \langle \nabla_\theta \ell(z, \theta_2), \theta_1 - \theta_2 \rangle + \frac{H}{2} \|\theta_1 - \theta_2\|_2^{\alpha+1}$.*

By the definition, it is easy to follow that if $\alpha = 1$, it is equivalent to $H$-smooth; and if $\alpha \to 0$, it satisfies the Lipschitz property given in Assumption 1. Besides, with bounded parameter space, i.e. $\|\mathcal{C}\|_2 \le M_\mathcal{C}$, $\alpha$-Hölder smoothness immediately implies $\max\{2HM_\mathcal{C}, H\}$-Lipschitz. Moreover, Assumption 3 instantiates many non-smooth loss functions. For example, the $q$-norm hinge loss $\ell(z, \theta) = (\max(0, 1 - y\langle \theta, z \rangle))^q$ for classification and the $q$-th power absolute distance loss $\ell(z, \theta) = |y - \langle \theta, z \rangle|^q$ for regression (Lei & Ying, 2020a), whose $\ell$ are $(q - 1)$-Hölder smooth if $q \in (1, 2]$ (Li & Liu, 2021). Lemma 1 shows that Hölder smoothness shares similar property with smoothness defined in Assumption 2, details of the proof can be found in Appendix A.1.

# 4 SHARPER UTILITY BOUNDS FOR DIFFERENTIALLY PRIVATE MODELS

## 4.1 PRIVACY GUARANTEES

Before analyzing the excess population risk bound, we first discuss the privacy guarantees in this section. Abadi et al. (2016) proposed the moments accountant method to measure the privacy costs of DP model training by stochastic gradient descent (SGD), Wang et al. (2017) further analyzed it under the setting of gradient descent (GD). In this paper, we focus more on the utility analysis, to improve the excess population risk, so we directly apply it to the gradient perturbation method.

**Lemma 2** (Wang et al. (2017)). *In gradient perturbation method in (1), for $\epsilon, \delta > 0$, it is $(\epsilon, \delta)$-DP if the random noise $b$ is zero mean Gaussian noise, i.e. $b \sim \mathcal{N}(0, \sigma^2 I_p)$, and for some constant $c$,*

$$\sigma^2 = c\frac{G^2 T \log(1/\delta)}{n^2 \epsilon^2}. \tag{3}$$

**Remark 1.** *(3) assumes the loss function to be $G$-Lipschitz. If we only assume that $\ell(\cdot, \cdot)$ is $\alpha$-Hölder smooth with parameter $H$, then $G$ can be replaced by $\max\{2HM_\mathcal{C}, H\}$ as discussed above.*

## 4.2 ANALYSIS OF THE EXCESS POPULATION RISK

To remove the $\mathcal{O}(1/\sqrt{n})$ term in previous results, we further need the Generalized Bernstein condition when analyzing the excess population risk.

**Assumption 4** (Generalized Bernstein condition (Koltchinskii, 2006)). *We say the loss function $\ell$ satisfies the generalized Bernstein condition if for some $B > 0$ for any $\theta \in \mathcal{C}$, we have:*

$$\mathbb{E}\left[(\ell(z, \theta) - \ell(z, \theta^*))^2\right] \le B(R(\theta) - R(\theta^*)).$$

Assumption 4 is a general condition, if the loss function $\ell(\cdot, \cdot)$ is $G$-Lipschitz and bounded by $M_\ell$, then many loss functions satisfy the generalized Bernstein condition, such as exponential loss function, logistic loss function, quadratic loss function, truncated quadratic loss, and hinge loss (Bartlett et al., 2006; Steinwart & Christmann, 2008).

Most of the previous works assumed that the loss function is convex (or strongly convex) when analyzing the optimization error (the excess empirical risk) $R_n(\hat{\theta}_n) - R_n(\theta_n^*)$. In this paper, we use the Polyak-Łojasiewicz (PL) condition to replace the convexity assumption.

**Assumption 5** (Polyak-Łojasiewicz condition). *The empirical risk $R_n(\theta)$ satisfies the Polyak-Łojasiewicz (PL) condition if there exists $\mu > 0$ and for every $\theta$,*

$$\|\nabla_\theta R_n(\theta)\|_2^2 \geq 2\mu \left(R_n(\theta) - R_n(\theta_n^*)\right).$$

The Polyak-Łojasiewicz condition is one of the weakest curvature conditions, so all the results given in this paper can be expanded to strongly convex conditions. (Karimi et al., 2016; Li & Liu, 2021), weaker than 'one-point convexity' (Kleinberg et al., 2018), 'star convexity' (Zhou et al., 2019), and 'quasar convexity' (Hinder et al., 2020). It is widely used in the analysis of non-convex learning (Wang et al., 2017; Charles & Papailiopoulos, 2018; Lei & Ying, 2020b; Lei & Tang, 2021) and many popular non-convex objective functions satisfy the PL condition, such as: matrix factorization (Liu et al., 2016), robust regression (Liu et al., 2016), neural networks with one hidden layer (Li & Yuan, 2017), mixture of two Gaussians (Balakrishnan et al., 2017), ResNets with linear activations (Hardt & Ma, 2017), linear dynamical systems (Hardt et al., 2018), phase retrieval (Sun et al., 2018), and blind deconvolution (Li et al., 2019).

**Remark 2.** *With $G$-Lipschitz and $\lambda$-strongly convex, we have $\mathbb{E}\left[\left(\ell(z,\theta) - \ell(z,\theta^*)\right)^2\right] \leq G^2 \|\theta - \theta^*\|_2^2$, and $R(\theta) - R(\theta^*) \geq \frac{\lambda}{2}\|\theta - \theta^*\|_2^2$, which implies $\mathbb{E}\left[\left(\ell(z,\theta) - \ell(z,\theta^*)\right)^2\right] \leq \left(2G^2/\lambda\right)\left(R(\theta) - R(\theta^*)\right)$, Assumption 4 is naturally satisfied. And PL condition can be directly derived from strongly convex (Karimi et al., 2016), so all strongly convex loss functions satisfy Assumptions 4 and 5 simultaneously and all the results given in this paper can be directly extended to strongly convex condition. Expect for strongly convex functions, several interesting machine learning setups also satisfy Assumptions 4 and 5. (1) 1-layer neural networks with a squared error loss and leaky ReLU activations. Charles & Papailiopoulos (2018) shows that 1-layer neural networks with a squared error loss and leaky ReLU activations satisfy Assumption 5, and Bartlett et al. (2006) shows that quadratic functions satisfy Assumption 4, so (1) holds. (2) Loss functions of least squares minimizations. Charles & Papailiopoulos (2018) shows that least squares minimization satisfy Assumption 5 and Bartlett et al. (2006) shows that the quadratic functions satisfy Assumption 4, so (2) holds. (3) Squared piecewise-linear functions with regularized term. Bartlett et al. (2006) shows that the composition of strongly convex functions with piecewise-linear functions satisfy Assumption 5, and Bartlett et al. (2006) shows that squared piecewise-linear functions satisfy Assumption 4. We prove that if a function satisfies Assumption 4, then with regularized term $\lambda\|\theta\|_2^2$, it also satisties Assumption 4 (details can be found in Appendix A.5). Thus, (3) holds.*

**Theorem 1.** *If Assumptions 1, 2, 4 and 5 hold, the loss function is bounded, i.e. $0 \leq \ell(\cdot,\cdot) \leq M_\ell$, taking $\sigma$ given by Lemma 2, $T = \mathcal{O}\left(\log(n)\right)$, $\eta_1 = \cdots = \eta_T = \frac{1}{L}$, if $\zeta \in (\exp(-p/8), 1)$, then with probability at least $1 - \zeta$:*

$$\begin{aligned}
R(\hat{\theta}_n) - R(\theta^*) \leq & \, c_1 \frac{G^2 p \log(n) \log(1/\delta)}{n^2 \epsilon^2} \left(1 + \left(\frac{8\log(T/\zeta)}{p}\right)^{1/4}\right)^2 \\
& + c_2 \left(\frac{G^2 \log^2(n)}{n} + \frac{B + M_\ell}{n}\right) \\
& + c_3 \frac{G^2 \log^{2.5}(n)\sqrt{p\log(1/\delta)}}{n\epsilon} \left(1 + \left(\frac{8\log(1/\zeta)}{p}\right)^{1/4}\right).
\end{aligned}$$

*for some constants $c_1, c_2, c_3 > 0$.*

Detailed proof can be found in Appendix A.2, we give a proof sketch here. First, we discuss the stability of the gradient perturbation based DP algorithm and show that it is $\mathcal{O}\left(T\eta/n\right)$ uniformly stable w.r.t $n$ with high probability. Then, we analyze the generalization error via stability theory. Meanwhile, via Assumption 4 and its moments bound, we couple term $R(\hat{\theta}_n) - R_n(\hat{\theta}_n)$ (the generalization error of $\hat{\theta}$) and term $R_n(\theta^*) - R(\theta^*)$ in (2) together, to remove the $\mathcal{O}\left(1/\sqrt{n}\right)$ term in the generalization error. In this way, a better excess population risk bound is achieved by combining the optimization error together.

The proof is motivated by Klochkov & Zhivotovskiy (2021) in the non-private case. The key challenges include that in the setting of DP, the random noise is injected into the algorithm. In Klochkov & Zhivotovskiy (2021), a key step to analyze the generalization error is summing

$X_i = \mathbb{E}'\left[\ell(z_i, \theta'_n) - \ell(z_i, \theta^*)\right]$ for $i = 1, \cdots, n$, where $\theta'_n$ is derived from an independent copy of the original dataset and $\mathbb{E}'$ means the expectation taken over the independent copy. When summing, $X_i$ is required to be zero mean. However, in the cases of DP, if we replace $\theta'_n$ by $\hat{\theta}'_n$, then $X_i$ are not zero mean. Besides, for output perturbation, a common way to decompose the excess population risk is $R(\hat{\theta}_n) - R(\theta^*) \leq R(\hat{\theta}_n) - R(\theta_n) + R(\theta_n) - R_n(\theta_n) + R_n(\theta_n) - R_n(\theta^*_n) + R_n(\theta^*) - R(\theta^*)$, which naturally solves the problem mentioned above (because the generalization error is discussed over the non-private model). However, when it comes to the gradient perturbation method, we cannot solve the problem easily in this way, because the random noise is coupled with the gradient. So, we decouple the noise terms and overcome the challenge by the moment Bernstein inequality.

By Theorem 1, it is easy to follow that with high probability, $R(\hat{\theta}_n) - R(\theta^*) = \mathcal{O}\left(\frac{\sqrt{p}}{n\epsilon}\right)$, which is the first $\mathcal{O}(1/n)$ high probability excess population risk bound over DP algorithm w.r.t $n$, to the best of our knowledge.

**Theorem 2.** *If Assumptions 3, 4, 5 hold, the loss function and the parameter space are bounded, i.e.* $0 \leq \ell(\cdot, \cdot) \leq M_\ell$, $\|\mathcal{C}\|_2 \leq M_\mathcal{C}$. *Taking $\sigma$ given by Lemma 2, $T = \mathcal{O}\left(n^{\frac{2}{1+2\alpha}}\right)$, and $\eta_t = \frac{2}{\mu(t+\kappa)}$, where $\kappa \geq \frac{2H^{1/\alpha}}{\mu}$, if $\zeta \in (\exp(-p/8), 1)$, then with probability at least $1 - \zeta$:*

$$
R(\hat{\theta}_n) - R(\theta^*) \leq c_1 \frac{G'^2 \sqrt{p \log(1/\delta)}}{n^{\frac{2\alpha}{1+2\alpha}} \epsilon} \left(1 + \left(\frac{8 \log(T/\zeta)}{p}\right)^{1/4}\right)
$$
$$
+ c_2 \left(\frac{G'^2 \log^2(n)}{n} + \frac{B + M_\ell}{n}\right)
$$
$$
+ c_3 \frac{G'^2 \log^2(n) \sqrt{p \log(1/\delta)}}{n^{\frac{2\alpha}{1+2\alpha}} \epsilon} \left(1 + \left(\frac{8 \log(1/\zeta)}{p}\right)^{1/4}\right).
$$

*for some constants $c_1, c_2, c_3 > 0$, where $G' = \max\{2HM_\mathcal{C}, H\}$.*

Detailed proof can be found in Appendix A.3. The proof is similar to Theorem 1, the challenge is that the properties $G$-Lipschitz and $L$-smooth are replaced by the assumption $\alpha$-Hölder smooth when analyzing the optimization error (the excess empirical risk). To overcome the challenge, we use Lemma 1 to bound the optimization error and Young's inequality is used to normalize the exponential rate, details are shown in the Appendix.

By Theorem 2, it is easy to follow that with high probability,

$$
R(\hat{\theta}_n) - R(\theta^*) = \mathcal{O}\left(\frac{\sqrt{p}}{\epsilon} n^{\frac{-2\alpha}{1+2\alpha}}\right).
$$

By the definition of $\alpha$-Hölder smooth, $\alpha \in (0, 1]$, so if $\alpha \in [\frac{1}{2}, 1]$,

$$
R(\hat{\theta}_n) - R(\theta^*) = \mathcal{O}\left(n^{\frac{-2\alpha}{1+2\alpha}}\right) \leq \mathcal{O}\left(n^{-\frac{1}{2}}\right)
$$

w.r.t $n$, which implies that our result is better than previous results when $\alpha \in [\frac{1}{2}, 1]$.

Via the discussion mentioned above, we observe that under the assumption $\alpha$-Hölder smooth, our result is better than $\mathcal{O}(1/\sqrt{n})$ w.r.t $n$ only in the case that $\alpha \in [\frac{1}{2}, 1]$. Besides, the best result is $\mathcal{O}\left(n^{-2/3}\right)$, which comes when $\alpha = 1$. And it cannot achieve the convergence rate $\mathcal{O}(\frac{\sqrt{p}}{n\epsilon})$. The reason is that when applying Young's inequality in the optimization error analysis, an additional term $\frac{H\eta_t^{\alpha+1}(1-\alpha)}{2(\alpha+1)}$ appears, leading a loose excess population risk bound.

Motivated by this, we design a variant of gradient perturbation method given in (1), called **max$\{1, g\}$-Normalized Gradient Perturbation** DP algorithm, to overcome the loose excess population risk bound. Details are shown in Algorithm 1.

**Remark 3.** *The difference between Algorithm 1 and (1) is that in lines 4 and 5, we normalize the $\ell_2$-norm of the gradient to $1$ if it is less than $1$. In this way, we can 'bypass' the Young's inequality when scaling $\|\theta_t - \theta^*_n\|_2^{1+\alpha}$ (derived from Lemma 1), further remove term $\frac{H\eta_t^{\alpha+1}(1-\alpha)}{2(\alpha+1)}$ in the theoretical analysis. Details can be found in Appendix A.4.*

---

**Algorithm 1** $\max\{\mathbf{1}, \mathbf{g}\}$-Normalized Gradient Perturbation

---

**Require:** dataset $D$, learning rate at iteration $t$: $\eta_t$, the variance of the Gaussian noise injected to the gradient: $\sigma$.

1: **function** M-NGP$(D, \eta_t, \sigma)$
2:      Initialize $\theta_0$.
3:      **for** $t = 0$ to $T - 1$ **do**
4:          **if** $\left\| \nabla_\theta R_n(\hat{\theta}_t) \right\|_2 < 1$ **then**
5:              $\nabla_\theta R_n(\hat{\theta}_t) \leftarrow \nabla_\theta R_n(\hat{\theta}_t) / \left\| \nabla_\theta R_n(\hat{\theta}_t) \right\|_2$.
6:          **endif**
7:          $\hat{\theta}_{t+1} \leftarrow \hat{\theta}_t - \eta_t \left( \nabla_\theta R_n(\hat{\theta}_t) + b \right)$, where $b \sim \mathcal{N}\left(0, \sigma^2 I_p\right)$.
8:      **endfor**
9:      return $\hat{\theta}_n = \hat{\theta}_T$.
10: **end function**

---

Then, via Algorithm 1, we can improve the excess population risk bound as shown below.

**Theorem 3.** *If Assumptions 3, 4, 5 hold, the loss function and the parameter space are bounded, i.e.* $0 \leq \ell(\cdot, \cdot) \leq M_\ell$, $\|\mathcal{C}\|_2 \leq M_{\mathcal{C}}$. *Taking $\sigma$ given by Lemma 2, $T = \mathcal{O}(\log(n))$, and $\eta_1 = \cdots = \eta_T = \eta$, where $\left( \frac{2}{H} - \frac{2^{-1/\alpha}}{\mu H^{(\alpha-1)/\alpha}} \right)^{1/\alpha} < \eta < \left( \frac{2}{H} \right)^{1/\alpha}$, if $\zeta \in (\exp(-p/8), 1)$, then with probability at least $1 - \zeta$,*

$$R(\hat{\theta}_n) - R(\theta^*) \leq c_1 \frac{G'\sqrt{p \log(n) \log(1/\delta)}}{n\epsilon} \left( 1 + \left( \frac{8 \log(T/\zeta)}{p} \right)^{1/4} \right)$$

$$+ c_2 \left( \frac{G'^2 \log^2(n)}{n} + \frac{B + M_\ell}{n} \right)$$

$$+ c_3 \frac{G'^2 \log^{2.5}(n) \sqrt{p \log(1/\delta)}}{n\epsilon} \left( 1 + \left( \frac{8 \log(1/\zeta)}{p} \right)^{1/4} \right),$$

*for some constants $c_1, c_2, c_3 > 0$, where $G' = \max\{2HM_{\mathcal{C}}, H\}$.*

Detailed proof can be found in Appendix A.4. The proof is similar to Theorems 1 and 2, the key difference is that by *gradient normalization* in Algorithm 1, Young's inequality is abandoned in the theoretical analysis (as discussed in Remark 3), which implies a better excess population risk bound.

By Theorem 3, it is easy to follow that with high probability, $R(\hat{\theta}_n) - R(\theta^*) = \mathcal{O}\left( \frac{\sqrt{p}}{n\epsilon} \right)$. The bound is of the same order as the result given in Theorem 1. This is also the first $\mathcal{O}(1/n)$ high probability excess population risk bound over DP algorithm w.r.t $n$ without smoothness assumption.

## 5 EXPERIMENTS

In this section, we perform experiments on real datasets to evaluate the difference between our proposed m-NGP algorithm and the traditional gradient perturbation (TGP), like (1).

The experiments are performed on classification task over datasets Iris (Dua & Graff, 2017), Breast Cancer (Mangasarian & Wolberg, 1990), Credit Card Fraud (Bontempi & Worldline, 2018), Bank (Moro et al., 2014), and Adult (Dua & Graff, 2017), the number of total data instances are 150, 699, 984, 41188, and 45222, respectively. We split the training and testing sets randomly and evaluate the accuracy on the testing set and the convengence rate on the training set. In all the experiments, the privacy budget $\delta$ is set $\frac{1}{n}$ and we choose $\epsilon = 0.1$ to 1.0.

We apply the regularized logistic regression method to the classification task, the loss function satisfies the assumptions mentioned before, and the experimental results are shown in Figure 1. We show the experimental results over datasets Iris and Adult in this section and experiments on other datasets are shown in Appendices B.1 and B.2. For convergence rate, the shadow area represents the

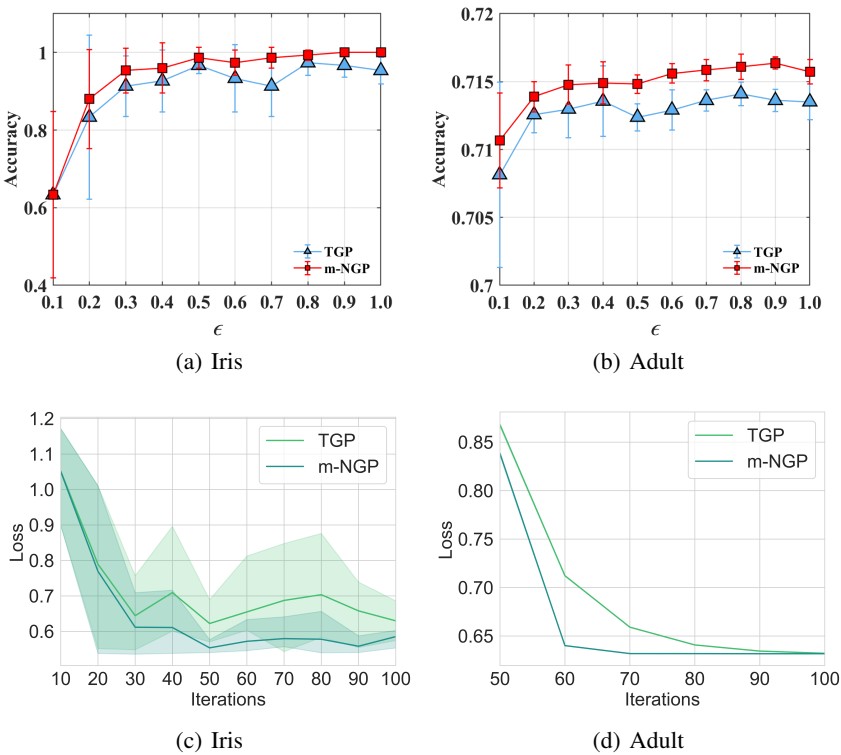

Figure 1: Comparisons between Traditional Gradient Perturbation (TGP) method and $\max\{\mathbf{1}, \mathbf{g}\}$-Normalized Gradient Perturbation (m-NGP) method.

maximum and minimum loss over mutiple experiments, reflecting the variance. The shadow area in part (d) of Figure 1 is not obvious, the reason is that the variances are small. Over most datasets, the accuracy and the convergence rate of $\max\{\mathbf{1}, \mathbf{g}\}$-Normalized Gradient Perturbation method is better than traditional gradient perturbation method. Besides, the accuracy of the DP model increases with the increasing of the privacy budget $\epsilon$, which is in line with the theoretical analysis.

## 6 CONCLUSIONS

In this paper, we first propose a state-of-the-art $\mathcal{O}\left(\frac{\sqrt{p}}{n\epsilon}\right)$ high probability excess population risk bound for gradient perturbation based DP algorithms, under the assumptions of $G$-Lipschitz, $L$-smooth, Polyak-Łojasiewicz condition, and generalized Bernstein condition. The result positively answers the open problem: *Can we achieve high probability excess risk bound with rate $\mathcal{O}(1/n)$ w.r.t $n$ for DP models via uniform stability?* Then, we extend the result to a more general case, requiring $\alpha$-Hölder smoothness, Polyak-Łojasiewicz condition, and generalized Bernstein condition. However, the result is not as satisfactory as before, we achieve an $\mathcal{O}\left(n^{\frac{-2\alpha}{1+2\alpha}}\right)$ high probability utility bound, which is better than previous results when $\alpha \in [\frac{1}{2}, 1]$ and cannot achieve an $\mathcal{O}(1/n)$ bound. To get a better result, we further propose a new algorithm: $\max\{1, g\}$-Normalized Gradient Perturbation (m-NGP). Detailed theoretical analysis shows that m-NGP can achieve $\mathcal{O}\left(\frac{\sqrt{p}}{n\epsilon}\right)$ high probability excess population risk bound, under the assumptions of $\alpha$-Hölder smoothness, Polyak-Łojasiewicz condition, and generalized Bernstein condition, which is the first $\mathcal{O}(1/n)$ high probability bound w.r.t $n$ under non-smoothness cases. Experimental results show that the accuracy of m-NGP algorithm is better than traditional gradient perturbation method. Thus, our proposed $\max\{1, g\}$-Normalized Gradient Perturbation method improves the excess population risk bound and the accuracy of the DP model over real datasets, simultaneously.

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
