# OpenReview forum: "Sharper Utility Bounds for Differentially Private Models"
_ICLR.cc/2022/Conference — ICLR 2022 Submitted_

### Official Review · Reviewer_9ZFZ · 2021-10-26

**Correctness:** 4
**Technical Novelty And Significance:** 3
**Empirical Novelty And Significance:** 1
**Recommendation:** 6
**Confidence:** 3

**Main Review:**

Strengths:

This paper is well-written and easy-to-follow. The presentation of results and proof sketches is clear.

The results of this paper are significant, improving an $O(1/\sqrt{n})$ term over previous utility bounds, with the help of introducing the Generalized Bernstein condition and the Polyak-Łojasiewicz condition.

Weakness:

The proof of Theorem 1 mainly follows Klochkov & Zhivotovskiy 2021, extending their result to the DP case using the same method. Theorem 2&3 have more novelty, but are less important than Theorem 1 in my opinion. The technical difficulties should be discussed in more details, to convince the readers that 1: the two new conditions are not too strong, 2: extending the results in Klochkov & Zhivotovskiy 2021 to the DP setting is non-trivial.

**Summary Of The Paper:**

This paper extends the results in Klochkov & Zhivotovskiy 2021 to the DP case, proving the first $O(1/n)$ high probability excess population risk bound for differentially private algorithms, assuming Lipschitzness, smoothness, Generalized Bernstein condition and Polyak-Łojasiewicz condition. Then the authors relax the assumptions of Lipschitzness+smoothness to $\alpha$-Holder smoothness, and propose a new normalized gradient pertubation algorithm that also achieves the $O(1/n)$ bound.

**Summary Of The Review:**

The presentation of results is clear. Extending existing results to the DP case and getting the first O(1/n) utility bound is significant. The method is mostly based on Klochkov & Zhivotovskiy 2021, thus not very novel.

---

> ### Author Response · Authors · 2021-11-17
> **Responses to Reviewer 9ZFZ**
>
> Thank you for the overall positive feedback and helpful comments.
>
> **Your Comment**: The technical difficulties should be discussed in more details, to convince the readers that the two new conditions are not too strong.
>
> **Our Response**: These two new conditions (Assumptions 4 and 5) are not too strong, there are many interesting machine learning setups satisfy them.
> As discussed in Section 4 (Corollary 1 and the footnote), for strongly convex loss functions (including but not limited to regularized logistic regression and mean squared error), Assumptions 4 and 5 can be naturally guaranteed. Except for the strongly convex loss function, the following examples also satisfy Assumptions 4 and 5. Firstly, Charles \& Papailiopoulos (2018) shows that 1-layer neural networks with a squared error loss and leaky ReLU activations satisfy the PL condition, and Bartlett et al. (2006) shows that squared piecewise-linear functions satisfies the Generalized Bernstein condition with parameter $2(M_\mathcal{C}+1)^2$, so 1-layer neural networks with a squared error loss and leaky ReLU activations satisfy Assumptions 4 and 5 simultaneously. Besides, Charles \& Papailiopoulos (2018) shows that least squares minimization satisfies the PL condition, and Bartlett et al. (2006) shows that the quadratic function satisfies the Generalized Bernstein condition with parameter $2(M_\mathcal{C}+1)^2$, so the loss functions of least squares minimizations also satisfy Assumptions 4 and 5 simultaneously. Moreover, Charles \& Papailiopoulos (2018) shows that the composition of strongly convex functions with piecewise-linear functions satisfy the PL condition, and Bartlett et al. (2006) shows that squared piecewise-linear functions satisfies the Generalized Bernstein condition with parameter $2(M_\mathcal{C}+1)^2$. In this paper, we furture show that squared piecewise-linear functions with regularized term $\lambda\\|\theta\\|_2^2$ also satisfy the Generalized Bernstein condition (Details can be found in Appendix A.5) and as a result, the squared piecewise-linear functions with regularized term satisfy Assumptions 4 and 5 simultaneously. To clarity this, we have added corresponding descriptions in the new version, details can be found in Remark 2.
>
> **Your Comment**: The technical difficulties should be discussed in more details, to convince the readers that extending the results in Klochkov \& Zhivotovskiy 2021 to the DP setting is non-trivial.
>
> **Our Response**: There is no doubt that we are inspired by Klochkov \& Zhivotovskiy (2021), however, when it comes to the setting of DP, especially for gradient perturbation method, we add random noise to each iteration when training, which brings technical difficulties to the theoretical analysis (not only for the generalization error, but also for the optimization error). In particular, for the generalization error, to keep the $\mathcal{O}(1/n)$ result overall under privacy setting (considering the random noise added to the gradient $T$ times) is difficult, as discussed in Theorem 1. Besides, for optimization error, extending previous results to the $\alpha$-H{\"older} smooth condition is also an important part, it gives similar results under more general cases. Meanwhile, from a theoretical point of view, we design a new algorithm to overcome the challenge brought by the Young's inequality, which may give some inspirations to other researchers when designing new gradient descent (or stochastic gradient descent) algorithms, with or without random noise.
>
> **References**
>
> [Charles \& Papailiopoulos (2018)] Zachary Charles and Dimitris Papailiopoulos. Stability and generalization of learning algorithms that converge to global optima. In Proceedings of the 35th International Conference on Machine Learning, pp. 745–754, 2018.
>
> [Bartlett et al. (2006)] Peter L Bartlett, Michael I Jordan, and Jon D McAuliffe. Convexity, classification, and risk bounds. Journal of the American Statistical Association, pp. 138–156, 2006.
>
> [Klochkov \& Zhivotovskiy (2021)] Yegor Klochkov and Nikita Zhivotovskiy. Stability and deviation optimal risk bounds with convergence rate $\mathcal{O}(1/n)$. arXiv preprint arXiv:2103.12024, 2021.

---

### Official Review · Reviewer_n773 · 2021-10-29

**Correctness:** 4
**Technical Novelty And Significance:** 4
**Empirical Novelty And Significance:** 3
**Recommendation:** 8
**Confidence:** 5

**Main Review:**

Strengths:
1. The problem is interesting and natural. This paper proposes the first $O(p^{0.5}/(n\epsilon))$ high probability excess population risk bound for DP algorithms under the assumptions Holder smooth, and PL condition (or Lipschitz, smooth, and PL condition). Comparing with previous results ($O(p^{0.5}/(n^{0.5}\epsilon))$), the improvement is significant, of the order $O(n^{0.5})$.

2. For the provided utility bounds, convexity of the loss function is replaced with the PL condition, so the results can be applied to many non-convex settings.

3. The paper also generalizes the $O(p^{0.5}/(n\epsilon))$ result to the non-smooth settings, by assuming the loss function to be Holder smooth and introducing normalization to traditional gradient perturbation method.

4. The results of this work are clear, novel concepts (techniques) are introduced to overcome the technical difficulties, and the technical contributions overall seem very solid. In this paper, the algorithmic stability is applied to bound the gap between the generalization error and its expectation, and via the Generalized Bernstein condition, the expectation one is solved. Another important part to achieve the superior theoretical results is to combining the generalization error together with term $R_n(\theta^*)$. Besides, the optimization error is solved under different assumptions, for smooth loss function, the proof is similar to previous works. But for non-smooth loss function, Holder smoothness is introduced and Young's inequality is applied to get the bound, and an additional term with $\eta$ exists. So the authors choose $\eta$ for each iteration carefully in Theorem 2 to get the claimed result. This is also the motivation of their proposed algorithm: to eliminate the additional term. After eliminating it, the proof of the new algorithm is similar to previous works.

5. Experiments on real datasets are performed to evalute their proposed algorithm m-NGP, the results show that the accuracy is better than traditional gradient perturbation DP method.

Weaknesses:
1. From my perspective, Holder smoothness implies $\max\{2HM_\mathcal{C},H\}$-Lipschitz if the parameter space is bounded by $M_\mathcal{C}$ (In Section 3, the authors claim it is $\max\{HM_\mathcal{C},H/2\}$-Lipschitz). The claim should be explained or corrected.

2. The paper overall read clearly, though the proof sketches should be more clear. For example, in Theorem 1, the authors say that 'the gradient perturbation based DP algorithm is $\mathcal{O}(T\eta/n)$ uniformly stable w.t.t n with high probability', but what are the connections between the uniformly stability and the given result? While the algorithmic stability is indeed closely related to the generalization error, the descriptions should be more detailed.

3. There are minor mistakes when analyzing the gap between $\theta_n$ and $\hat{\theta}_n$, although it does not affects the results signifacantly, the mistakes should be corrected.

4. Some of the notations should be defined before used. For example, $R_n(\hat{\theta}_{t-1})$ in (1) is formally defined after it, and $\eta_t$ in (1) is not explained throughout the paper. The authors should check them carefully.

5. Although the understandability of the current manuscript is acceptable, I encourage the authors to correct the typos and the grammatical errors in future versions.

**Summary Of The Paper:**

This paper analyzes the utility bounds of the gradient-perturbation based DP algorithm. They first provide DP by previous result (it is not the key point in this paper). Then, by applying the Generalized Bernstein condition, they give $O(p^{0.5}/(n\epsilon))$ high probability excess population risk bound under the properties Lipschitzness, smoothness, and PL condition. Furthermore, under the more general assumption (Holder smoothness), they analyze the utility bound but the result is not so good as before. So they propose an algorithm (called m-NGP) to improve it under the assumption Holder smooth, and it is claimed that the utility bound can be improved to $O(p^{0.5}/(n\epsilon))$. Their results are sharper than previous analyses in different settings: previous results require convex assumption but this paper requires PL condition. Moreover, experiments are performed to evaluate the accuracy of m-NGP.

**Summary Of The Review:**

The results given by this paper are clear, the theoretical analysis seems solid, the organization and the presentation are fine. But the proof sketches should be more clear, the notations, the typos, and the grammatical errors should be checked. Although there are some minor issues, I recommend this paper to be accepted because of the superior theoretical results and the novel technologies.

---

> ### Author Response · Authors · 2021-11-17
> **Responses to Reviewer n773**
>
> Thank you for the overall positive feedback and helpful comments.
>
> **Your Comment**: From my perspective, Holder smoothness implies $\max\\{2HM\_\mathcal{C},H\\}$-Lipschitz if the parameter space is bounded by $M\_\mathcal{C}$ (In Section 3, the authors claim it is $\max\\{HM\_\mathcal{C},H/2\\}$-Lipschitz). The claim should be explained or corrected.
>
> **Our Response**: We have corrected the Lipschitz constant descripted in Section 3 in the new version.
>
> **Your Comment**: The paper overall read clearly, though the proof sketches should be more clear. For example, in Theorem 1, the authors say that 'the gradient perturbation based DP algorithm is $\mathcal{O}(T\eta/n)$ uniformly stable w.t.t n with high probability', but what are the connections between the uniformly stability and the given result? While the algorithmic stability is indeed closely related to the generalization error, the descriptions should be more detailed.
>
> **Our Response**: We have added descriptions 'We analyze the generalization error via stability theory' and 'In this way, a better excess population risk bound is achieved by combing the optimization error together' after Theorem 1. Details can be found in Section 4.
>
> **Your Comment**: There are minor mistakes when analyzing the gap between $\theta\_n$ and $\hat{\theta}\_n$, although it does not affects the results signifacantly, the mistakes should be corrected.
>
> **Our Response**: To correct the mistake, we do not divide $R\_n(\hat{\theta}\_n)-R\_n(\theta^\*\_n)$ into $R\_n(\hat{\theta}\_n)-R\_n(\theta\_n)$ and $R\_n(\theta\_n)-R_n(\theta^\*\_n)$. In the new version, we directly discuss $R\_n(\hat{\theta}\_{t+1})-R\_n(\hat{\theta}\_{t})$ at each iteration $t$, and summing over $T$ iterations to get the result. Details can be found in Appendix A.2 (the proof of Lemmas 7 and 8) and Appendix A.4, in the new version.
>
> **Your Comment**: Some of the notations should be defined before used. For example, $R\_n(\hat{\theta}\_{t-1})$ in (1) is formally defined after it, and $\eta\_t$ in (1) is not explained throughout the paper. The authors should check them carefully.
>
> **Our Response**: We have carefully checked the notations to ensure symbols defined before used in the new version.
>
> **Your Comment**: Although the understandability of the current manuscript is acceptable, I encourage the authors to correct the typos and the grammatical errors in future versions.
>
> **Our Response**: We have tried our best to correct the typos and the grammatical errors in the new version.

---

> > ### Comment · Reviewer_n773 · 2021-11-19
> > **Thanks for the response**
> >
> > Thanks for the detailed responses from the authors.
> >
> > Other reviewers have asked many questions with high quality, such as to give examples satisfy Assumptions 4 and 5, and to give experiments to evaluate the claimed theorems. I have carefully read the comments and the responses, I think the authors answered these questions well and experimental results are in line with the theoretical analysis. Generally, to get a better utility bound, more assumptions are acceptable (such as Assumptions 4 and 5 in this paper), and the authors gave several loss functions that satisfy these two new conditions to demonstrate that they are general. I am familiar with this field, and have attempted to achieve a better high probability utility bound. This paper solves this problem elegantly without a too much complicated proof, and the proof techniques used in this paper may give improtant inspirations to other researchers (at least me).
> >
> > I am not sure that I am going to increase my already fairly high rating. But I am even more convinced that this submission should be accepted.

---

> > > ### Author Response · Authors · 2021-11-20
> > > **Thanks for the reply**
> > >
> > > Thanks for your highly recommendations. We are glad that you found the contributions of our work. Thanks again for your positive feedback and helpful comments.

---

### Official Review · Reviewer_AguJ · 2021-10-29

**Correctness:** 3
**Technical Novelty And Significance:** 3
**Empirical Novelty And Significance:** 3
**Recommendation:** 5
**Confidence:** 4

**Main Review:**

I think the problem statement of the paper is interesting and important. The authors aim to solve the important open problem in DP community: to achieve high probability excess population risk bounds with tighter upper bound. The paper is well-written and has clear writing logical.  The authors make solid theoretical contribution and based on the logical in theoretical analysis, they propose the new algorithm to achieve better excess risk bound. It would be good to bring the experiments on real datasets to verify theoretical results. But there are some limitations in the paper. Firstly, the authors do not give existence of function under Assumption 4 and Assumption 5 simultaneously. Secondly, the authors only give the upper bound of excess risk bound and there is no lower bound, so it is unknown that their bound is optimal. Thirdly, in experiment part, the authors do not descript the loss function task and if the function in experiments satisfies these assumptions in theoretical part. Besides, they should give the performance about changing privacy parameter and dimension parameter.

**Summary Of The Paper:**

The paper achieves high probability excess risk bound with rate O(1/n) w.r.t n for DP models via uniform stability by using Generalized Bernstein condition under G-Lipschitz, L-smooth, and PL condition. Then the authors expand the result to a more general case, only requiring α-Ho ̈lder smoothness, Polyak-Łojasiewicz condition, and generalized Bernstein condition. But the result is worse than before, so in order to get a better result, they propose m-NGP algorithm to achieve O(1/n) high probability bound w.r.t n under α-Ho ̈lder smoothness, Polyak-Łojasiewicz condition, and generalized Bernstein condition. The authors also show the experimental better accuracy results of m-NGP compared to traditional gradient perturbation method on real datasets.

**Summary Of The Review:**

It is not sure that there is the function satisfying assumption 4 and assumption 5,so the problem setting may be not reasonable. And the authors do not give the loss function task in detail in experimental part. There is no lower bound so we do not know if this upper bound is optimal. And the authors should give more experiments about different parameter.

---

> ### Author Response · Authors · 2021-11-17
> **Responses to Reviewer AguJ**
>
> Thank you for the overall positive feedback and helpful comments.
>
> **Your Comment**: It is not sure that there is the function satisfying assumption 4 and assumption 5,so the problem setting may be not reasonable.
>
> **Our Response**: As discussed in Section 4 (Corollary 1 and the footnote in the old version), for strongly convex loss functions (including but not limited to regularized logistic regression and mean squared error), Assumptions 4 and 5 can be naturally guaranteed. Except for the strongly convex loss function, the following examples show that several interesting machine learning setups satisfy Assumptions 4 and 5. Firstly, Charles \& Papailiopoulos (2018) shows that 1-layer neural networks with a squared error loss and leaky ReLU activations satisfy the PL condition, and Bartlett et al. (2006) shows that squared piecewise-linear functions satisfies the Generalized Bernstein condition with parameter $2(M_\mathcal{C}+1)^2$, so 1-layer neural networks with a squared error loss and leaky ReLU activations satisfy Assumptions 4 and 5 simultaneously. Besides, Charles \& Papailiopoulos (2018) shows that least squares minimization satisfies the PL condition, and Bartlett et al. (2006) shows that the quadratic function satisfies the Generalized Bernstein condition with parameter $2(M_\mathcal{C}+1)^2$, so the loss functions of least squares minimizations also satisfy Assumptions 4 and 5 simultaneously. Moreover, Charles \& Papailiopoulos (2018) shows that the composition of strongly convex functions with piecewise-linear functions satisfy the PL condition, and Bartlett et al. (2006) shows that squared piecewise-linear functions satisfies the Generalized Bernstein condition with parameter $2(M_\mathcal{C}+1)^2$. In this paper, we furture show that squared piecewise-linear functions with regularized term $\lambda\\|\theta\\|_2^2$ also satisfy the Generalized Bernstein condition. Combining the results together, it can be achieved that the squared piecewise-linear functions with regularized term satisfy Assumptions 4 and 5 simultaneously. To clarity this, we have added corresponding descriptions in the new version, details can be found in Remark 2 and Appendix A.5.
>
> **Your Comment**: And the authors do not give the loss function task in detail in experimental part.
>
> **Our Response**: In the experiments shown in the main text, the loss function is regularized logistic regression, which satisfies all the assumptions in the theoretical part. To clarify this, we describe it in the new version, details can be found in Section 5.
>
> **Your Comment**: There is no lower bound so we do not know if this upper bound is optimal.
>
> **Our Response**: In this paper, we focus on the upper bound of the excess population risk. However, the lower bound is an important problem. We are considering about this problem under these assumptions, but have no rigorous proofs till now. We will study the lower bound and discuss whether the upper bound given in this paper is the optimal one in our future work.
>
> **Your Comment**: And the authors should give more experiments about different parameter.
>
> **Our Response**: In the new version, experiments over more $\epsilon$ are added, and we also perform experiments to evaluate the effects caused by dimension parameter $p$. The results show that with the increasing of $\epsilon$, the accuracy becomes better, and the result of m-NGP is better than TGP on most datasets (details can be found in Section 5 and Appendix B.1). Besides, to evaluate how $p$ effect the accuracy, we append dimensions filled by 0 to the original data (to make $p$ larger without introducing or reducing new information). The experimental results show that with the increasing of $p$, the accuracy decreases overall, which is in line with the theoretical analysis. Details can be found in Appendix B.3.
>
>
> **References**
>
> [Charles \& Papailiopoulos (2018)] Zachary Charles and Dimitris Papailiopoulos. Stability and generalization of learning algorithms that converge to global optima. In Proceedings of the 35th International Conference on Machine Learning, pp. 745–754, 2018.
>
> [Bartlett et al. (2006)] Peter L Bartlett, Michael I Jordan, and Jon D McAuliffe. Convexity, classification, and risk bounds. Journal of the American Statistical Association, pp. 138–156, 2006.

---

> > ### Comment · Reviewer_AguJ · 2021-12-04
> > **Response**
> >
> > For logistic regression, actually it is also generalized linear whose error bound can be independent on the dimension, so the bound in this paper is only suboptimal. Thus, this is also a concern in the experimental part.
> > For  squared piecewise-linear functions and 1-layer nn with ReLU they are non-smooth, right? And your paper needs it to be a smooth loss. Thus, it seems like only least squares minimization satisfies the assumption in the paper. However, it is still quite limited.

---

> > > ### Author Response · Authors · 2021-12-04
> > > **Responses to Reviewer AguJ**
> > >
> > > To Reviewer AguJ:
> > >
> > > Thank you for the feedback and we will clarify your concerns in the following.
> > >
> > > Firstly, one of the key parts of our paper is to expend the smoothness assumption to H{\"o}lder smoothness.
> > > And based on theoretical analysis, we further design a new algorithm, and the new algorithm achieves the same utility bound as under smooth and Lipschitz conditions.
> > > Thus, our proposed Theorems 2 and 3 only require the loss function to be H{\"o}lder smooth, rather than smooth.
> > > As discussed in Section 3, the $q$-norm hinge loss and $q$-th power absolute distance loss can be seemed as squared piecewise-linear loss functions when $q=2$, so they satisfy the assumption H{\"o}lder smooth.
> > > For one-layer neural networks with squared error loss and leaky ReLU activations, the same phenomenon holds because the neural network can be seemed as a matrix multiplication.
> > > It is a misunderstanding that the results given in this paper only cover logistic regression and simple least squared minimization.
> > > We give examples in the following: (1) logistic regression models; (2) least squared minimizations; (3) $q$-norm hinge loss when $q=2$; (4) one-layer neural networks with squared error loss and leaky ReLU activations.
> > > Besides, the examples given in this paper and listed above are only part of the loss functions who satisfy those assumptions.
> > > We do not only focus on simple least squared minimizations and the logistic regression model, which have been discussed in detail by previous works, but extend the condition from smoothness to H{\"o}lder smoothness, and from strongly convex (convex) to the PL condition (some of the non-convex conditions).
> > >
> > > Secondly, for DP logistic regression model, one of the results we know is given by Chaudhuri et al. (2011), which indeed gave an $\mathcal{O}\big(\frac{1}{\sqrt{n}}\big)$ high probability excess population risk bound w.r.t $n$, for objective and output perturbation methods. The result is independent on $p$.
> > > However, we give an $\mathcal{O}\left(\sqrt{p}/n\right)$ high probability bound w.r.t $n,p$ in this paper.
> > > When it comes to the condition that $p<n$ (which is common in the field of machine learning), our method is superior.
> > >
> > > We hope that our response will eliminate you concerns and if there exists any further questions, we are glad to clarity them in detail.
> > >
> > > References:
> > >
> > > [Chaudhuri et al. (2011)] Kamalika Chaudhuri, Claire Monteleoni, and Anand D Sarwate. Differentially private empirical risk minimization. Journal of Machine Learning Research, pp. 1069–1109, 2011.

---

### Official Review · Reviewer_wceT · 2021-11-02

**Correctness:** 3
**Technical Novelty And Significance:** 2
**Empirical Novelty And Significance:** 2
**Recommendation:** 5
**Confidence:** 4

**Main Review:**

Pros:
The authors discussed each assumption in detail, which gives the result under various assumptions.
The analysis hints that the generalized Bernstein inequality could provide a new tool for improving population risk bound and other differentially private algorithms.
The algorithm is empirically evaluated on a number of datasets and achieved noteworthy results.

Questions:
It seems that mNGP is proposed solely for the sake of bypassing Young’s inequality. However in the empirical evaluation it seems that the normalization works well on various datasets. Could the authors elaborate on this and provide some intuition on why normalization works in practice?
For the experiment results, could you report the variance and the number of random seeds used? Will it be open source?
The paper shows that normalization helps with improving population risk bound, though the convergence rate of the algorithm is yet to be discussed. Could the authors provide an insight on whether the normalization will affect the convergence rate? If so, to what extent should we expect normalization to affect the rate?

Technical questions:
For Theorem 3, in the step where you get R_n(\hat{\theta}_t) - R_n(\theta^\ast_n), the authors refer to Lemma 7, which gives us the bound on R_n(\theta_t) - R_n(\theta^\ast_n). This along with R_n(\theta_n) - R_n(\theta^\ast_n) does not give a bound for R_n(\hat{\theta}_n) - R_n(\theta^\ast_n). How did it go through?
In the proof of Lemma 3, why do we need to assume $i=n$? This condition is not used in (4) and the remaining proofs.
Bousquet lemma is referred both as lemma 4 and lemma 7 in the section.
Appendix A.2 Lemma 3: Consider stating lemma 4 first and then state and prove lemma 3 for the sake of clarity. The same goes for Lemma 5 and 6. It is also unclear to me why ``Bounding’’ is in bold text.
• For (17), is \psi is defined prior to this equation?

**Summary Of The Paper:**

This paper gives a high probability excess population risk bound for differentially private optimization algorithm under $G$-Lipschitz, $L$-smooth and PL condition with gradient perturbation. This paper also gives a result under $\alpha$ Holder smoothness, with an optimized bound and a new normalized gradient perturbation scheme.

**Summary Of The Review:**

This manuscript provides a sharper utility analysis up to some clarity in the proofs. There's also some reservation on the significance of the techniques presented in the proofs.

---

> ### Author Response · Authors · 2021-11-17
> **Responses to Reviewer wceT**
>
> Thank you for the overall positive feedback and helpful comments.
>
> **Your Comment**: It seems that mNGP is proposed solely for bypassing Young’s inequality. However the normalization works well on various datasets. Could the authors elaborate on this and provide some intuition on why normalization works in practice?
>
> **Our Response**: The normalization scales the $\ell_2$-norm of the gradient to 1 if it is less than 1, strengthens the gradient. The intuition is that in the setting of gradient perturbation, noise is injected into the gradient, when the gradient is small, the random noise will play a more improtant role and the gradient property will be concealed beneath the noise. So, strengthening the gradient may prevent this phenomenon. Besides, from the theoretical view, we prove that better utility bounds are achieved by introducing normalization; and experiments also show that m-NGP enhances the accuracy and the convergence rate (detailed in Section 5 and Appendices B.1, B.2 in the new version).
>
> **Your Comment**: For experiments, could you report the variance and the number of random seeds? Will it be open source?
>
> **Our Response**: We perform new experiments in the new version, showing variances, and the random seed is set 10. Besides, we perform experiments over more $\epsilon$. The results show that the accuracy of  m-NGP is better than TGP. Details can be found in Section 5 and Appendix B.1 in the new version. And the code is given in the supplementary material.
>
> **Your Comment**: The convergence rate of the algorithm is yet to be discussed. Could the authors provide an insight on whether the normalization will affect the convergence rate? If so, to what extent?
>
> **Our Response**: As answered in Comment 1, normalization strengthens the gradient, prevents the random noise to play a more important role when training. Besides, from theoretical perspective, normalization improves the convengence rate from $\mathcal{O}(n^{\frac{2}{1+2\alpha}})$ to $\mathcal{O}(\log(n))$, and a better utility bound is achieved. Moreover, in the new version, we have performed experiments to show the effects on the convergence rate caused by normalization, the results show that m-NGP achieves faster convergence rate. Details can be found in Section 5 and Appendix B.2 in the new version.
>
> **Your Comment**: For Theorem 3, when you get $R\_n(\hat{\theta}\_t)-R\_n(\theta^\*\_n)$, the authors refer to Lemma 7, which gives us the bound on $R\_n(\theta\_t)-R\_n(\theta^\*\_n)$. This along with $R\_n(\theta\_n)-R\_n(\theta^\*\_n)$ does not give a bound for $R\_n(\hat{\theta}\_n)-R_n(\theta^\*\_n)$. How did it go through?
>
> **Our Response**: The reviewer may misunderstand the proof process in the old version. Actually, we first analyze $R\_n(\hat{\theta}\_n)-R\_n(\theta\_n)$, and then analyze $R\_n(\theta\_T)-R_n(\theta^\*\_n)$, considering that $\theta\_n$ is the same as $\theta\_T$, we get $R\_n(\hat{\theta}\_n)-R\_n(\theta^\*\_n)$ by summing them. To clarify the proof process, in the new version, we do not divide $R\_n(\hat{\theta}\_n)-R\_n(\theta^\*\_n)$ into these two terms, and directly discuss $R\_n(\hat{\theta}\_{t+1})-R\_n(\hat{\theta}\_t)$ at iteration $t$. Considering that $\hat{\theta}\_n$ is the same as $\hat{\theta}\_T$, we get the result after $T$ iterations. Details can be found in Appendix A.2 (the proof of Lemmas 7 and 8) and Appendix A.4, in the new version.
>
> **Your Comment**: In the proof of Lemma 3, why do we need to assume $i = n$? This condition is not used in (4) and the remaining proofs.
>
> **Our Response**: Actually it is not necessary to assume $i=n$, the assumption is only for the sake of simplicity, the result holds for every $i\in[1,\cdots,n]$. This is also the reason that 'this condition is not used in (4) and remaining proofs', we have removed this assumption in the new version for clarity (see the proof of Lemma 5 in the new version).
>
> **Your Comment**: Bousquet lemma is referred both as lemma 4 and lemma 7.
>
> **Our Response**: Lemma 4 in this paper and Lemma 7 in Bousquet et. al. (2020) are two different lemmas, we add lemma 7 in Bousquet et. al. (2020) to the new version for clarity (see Lemma 4 in the new version).
>
> **Your Comment**: Consider stating lemma 4 first and then state and prove lemma 3 for the sake of clarity. The same goes for Lemma 5 and 6.
>
> **Our Response**: In the new version, we state Lemmas 3 (Lemma 4 in the old version), 4 before Lemma 5 (Lemma 3 in the old version) and state Lemma 6 (Lemma 6 in the old version) before Lemma 7 (Lemma 5 in the old version) for the sake of clarity.
>
> **Your Comment**: It is unclear why `Bounding' is in bold text.
>
> **Our Response**: We have removed the bold text of 'Bounding' in the appendix in the new version.
>
> **Your Comment**: For (17), is $\varphi$ is defined prior to this equation?
>
> **Our Response**: For (17), $\varphi$ can be any value larger than 0 (we explain it below inequality (17)), and it can be seemed as an constant in this section.

---

> > ### Comment · Reviewer_wceT · 2021-11-19
> > **Thanks for the response**
> >
> > I've read the response and most of my questions have been addressed. The arguments in the paper look sound to me. The technical contribution of the paper remains to be seemingly limited to me. With the authors' response, I am not opposed to an acceptance.

---

> > > ### Author Response · Authors · 2021-11-20
> > > **Thanks for the reply**
> > >
> > > Thanks for your reply and we are glad our answers helped address your concerns. We are grateful for you recognitions of our work. For the technical contribution, as pointed out by Reviewer n773, we get the sharper high probablity bound via a not too complicated, but elegant approach. Thanks again for your insightful feedback and helpful comments.

---

### Author Response · Authors · 2021-11-17
**General Response**

We sincerely thank all the referees for their appreciation of our work and providing highly constructive comments for improvement. We have carefully revised the manuscript based on the initial reviews. The following is a summary of major changes:

* In Remark 2 and Appendix A.5, we discuss the loss functions which satisfy Assumptions 4 (generalized Bernstein condition) and 5 (PL condition) in detail.
* In Section 5 and Appendix B.1, we give the experimental results over more privacy parameters $\epsilon$, from 0.1 to 1.0, the variances taken over multiple experiments are also given.
* In Section 5 and Appendix B.2, we add experiments to evaluate the effects on the convegence rate caused by m-NGP.
* In Appendix B.3, we add experiments to evaluate the effects on the accuracy caused by different dimension parameters $p$.
* After Theorem 1, we give more details of the proof sketch.

We hope that the given concerns have been addressed satisfactorily in the revised manuscript and the point-by-point responses to the reviewers' comments.

---

> ### Author Response · Authors · 2021-11-27
> **Thanks for great efforts from AC and reviewers**
>
> Dear AC and reviewers:
>
> Thank you again for the great efforts and the valuable comments. We have carefully addressed the main concerns in detail. We hope you might find the responses satisfactory. As the discussion phase is about to close, we are very much looking forward to hearing from you about any further feedback. We will be very happy to clarify any further concerns (if any).
>
> Best,
>
> Authors.

---

### Decision · Program_Chairs · 2022-01-20

**Decision:**

Reject

**Comment:**

The paper gives high probability bounds on excess risk for differentially private learning algorithms, in the setting where the loss is assumed to be Lipschitz, smooth, and assumed to satisfy the Polyak-Łojasiewicz (PL) condition. The key idea in the paper is to leverage the curvature in the loss (PL condition) and the generalized Bernstein condition.

Authors show that they get sharper bounds of the order \sqrt{p}/(n\epsilon) when the loss is assumed to satisfy the PL condition besides being convex Lipschitz/smooth. Without using some curvature information about the loss function, the best upper bounds we can get are in the order of \sqrt{p}/(n\epsilon) + 1/\sqrt{n} — and this is tight at least in terms of the dependence on n given the nearly matching lower bounds — in fact, the dependence on n is tight as it matches the non-private settings.

So, I find it a bit misleading when authors say that they improve over the existing results. That statement is not true in its generality — it is true that we can leverage the PL condition to give faster rates but that is not the setting of prior work. Again, the bounds that authors compare against are for smooth/Lipschitz convex loss functions and without any assumption on the curvature of the loss.

If we do look at the literature for when and/or how can curvature help, we can compare against the existing bounds for strongly convex losses. The best-known result in the setting that is most closely related is that of Feldman et al. (STOC 2020): https://dl.acm.org/doi/pdf/10.1145/3357713.3384335. As we can check from Theorem 4.9 in that paper, the bounds we get are in the order of 1/n + d/n^2 which is actually better — not surprising since PL condition is a weaker condition. There is merit to the results in this paper but the current narrative is quite misleading and a more careful comparison with the existing literature is needed. The bounds are hard to parse — for example, what is the dependence on the strong convexity parameter (\mu)? It would also help to instantiate specific loss functions so that we can fix some of the parameters in the bound to have a clear comparison with the existing bounds.